# Anti-Inflammatory Activity of APPA (Apocynin and Paeonol) in Human Articular Chondrocytes

**DOI:** 10.3390/ph17010118

**Published:** 2024-01-16

**Authors:** Mercedes Fernández-Moreno, Tamara Hermida-Gómez, Nicholas Larkins, Alan Reynolds, Francisco J. Blanco

**Affiliations:** 1Grupo de Investigación en Reumatología (GIR), Instituto de Investigación Biomédica de A Coruña (INIBIC), Complexo Hospitalario Universitario de A Coruña (CHUAC), Sergas, Universidade de A Coruña (UDC), 15071 A Coruña, Spain; tamara.hermida.gomez@sergas.es; 2Grupo de Investigación en Reumatología y Salud (GIR-S), Centro Interdisciplinar de Química y Biología (CICA), Universidade de A Coruña (UDC), Campus de Elviña, 15071 A Coruña, Spain; 3Centro de Investigación Biomédica en Red, Bioingenieria, Biomatereial y Nanomedicina (CIBER-BBN), 50018 Zaragoza, Spain; 4AKL Therapeutics Ltd., Stevenage Bioscience, Gunnels Wood Rd, Stevenage SG1 2FX, UK; nl@aklrd.com (N.L.); ar@aklrd.com (A.R.); 5Grupo de Investigación en Reumatología y Salud (GIR-S), Departamento de Fisioterapia, Medicina y Ciencias Biomédicas, Facultad de Fisioterapia, Centro Interdisciplinar de Química y Biología (CICA), INIBIC-Sergas, Universidade de A Coruña (UDC), Campus de Oza, 15008 A Coruña, Spain

**Keywords:** APPA, human chondrocytes, OA, anti-inflammatory, ROS production

## Abstract

Osteoarthritis (OA) is a chronic joint disease leading to cartilage loss and reduction in the joint space which results in pain. The current pharmacological treatment of OA is inadequate and pharmacological interventions focus on symptom management. APPA, a combination of apocynin (AP) and paeonol (PA), is a potential drug for treating OA. The aim of this study was to analyze the effects of APPA on the modulation of the inflammatory response in chondrocytes. Samples were incubated with IL-1β and APPA, and their responses to proinflammatory cytokines, catabolic mediators and redox responses were then measured. The effect of APPA on mitogenesis was also evaluated. Results show that APPA attenuated the expression of IL-8, TNF-α, MMP-3, MMP-13, SOD-2 and iNOS, resulting in the protection of human articular cartilage. APPA decreased PGC-1α gene expression induced by IL-1β. APPA did not modulate the gene expression of Mfn2, Sirt-1 or Sirt-3. The overall findings indicate that APPA may be an effective treatment for OA by targeting several of the pathways involved in OA pathogenesis.

## 1. Introduction 

Osteoarthritis (OA) is a chronic joint disease leading to cartilage loss and reduction in the joint space, which results in pain. Pain is the dominant symptom of OA, and there is a significant unmet need for an effective treatment [1]. The global impact of OA is predicted to increase as obesity and longevity increase. Osteoarthritis is associated with increased mortality and decreased patient quality of life and places a significant burden on society [2]. Using data from the 2019 Global Burden of Diseases (GBD) study, it has been estimated that there are 398 million prevalent cases of hip and knee OA [3]. 

The current pharmacological treatment of OA is inadequate, and guidelines for OA advocate non-pharmacologic treatments as the core of initial interventions, including exercise, weight loss and education. This is followed by pharmacological interventions which focus on symptom management. The most widely used agents include oral and topical non-steroidal anti-inflammatory drugs (NSAIDs), acetaminophen and minor opioids [4]. Despite the importance of OA on patient QoL, the human cost of pain and disability caused and the economic costs, treatment options are limited. Due to a lack of disease-modifying therapy, patients with OA typically undergo joint replacement surgery at the end stage of the disease after experiencing joint pain and stiffness for decades. Pain is the dominant symptom of OA, and currently treatments are inadequate, resulting in a significant unmet need [1].

Recent research has established that OA involves the whole joint with not only a loss of cartilage but also changes in the subchondral bone, synovium, tendons, ligaments and muscles. It is now recognized that the disease process is more complex than originally assumed and, to a lesser or greater degree, involves chronic inflammation [5], mainly involving the innate immune system [6], triggered for example, by aging or obesity [7]. Aging itself has important consequences with regard to OA, including low grade inflammation (inflamm-aging), mitochondrial dysfunction with oxidative stress and cell senescence [8,9]. The roles of bone, cartilage and synovium in OA with cross-talk between them involve many different pathways and provide numerous potential treatment targets [10,11,12]. 

Historically, OA was generally accepted to be simply an age or mechanical stress-related degenerative joint disease; however, nowadays this concept is no longer sufficient to explain the entire process [13]. The pathological changes in all of the joint tissues (cartilage, subchondral bone, synovium, ligaments, muscle and menisci) are the impetus for considering OA as a disease and the joint as an organ [14]. With this greater understanding of OA and that the joint acts as an organ, this disease could be considered a systemic illness. OA has now been characterized by a localized loss of cartilage, the remodeling of adjacent bone and associated inflammation. The low but chronic grade of inflammation is clearly relevant to the development of new treatments [15]. In addition to inflammation, mitochondrial dysfunction is well documented in OA and has the capacity to alter chondrocyte function and viability, contributing to cartilage degradation [16,17,18,19,20,21].

Although chondroprotection has been a major focus of research, no disease modifying drugs have been approved for the treatment of OA. Disease-modifying OA drugs (DMOADs) have targeted the structural change of the disease by aiming to inhibit cartilage-degrading factors, promote cartilage production or target bone remodeling, inflammation or cell senescence, with or without effects on symptoms [8,22]. In recent years, the application of small molecule drugs to treat osteoarthritis by regulating several pathways related to OA pathogenesis have been reported in the literature, especially the research progress related to natural drugs and derivatives, which have the potential to effectively inhibit symptoms and regulate homeostasis [23,24,25,26,27,28].

APPA is a combination of two synthetic isomers of secondary plant metabolites apocynin (AP) and paeonol (PA). AP [29,30] and PA [31] have been reported to have a wide range of effects in animal models, tissue explants and cell systems across a range of diseases. AP, an NADPH oxidase inhibitor, is a strong ROS scavenger and inhibits the expression and release of several inflammatory cytokines and matrix metalloproteinases [31,32,33,34], while both PA and AP downregulate activation, nuclear translocation and the DNA binding of NF-κB [35,36,37]. In addition, PA has been reported to upregulate the gene expression of Nrf2, a key component of the response to oxidative stress [38]. The anti-inflammatory potential of APPA has been described in neutrophils [39]. APPA has been shown to provide pain relief in two studies in dogs with naturally occurring OA [40,41] and provided chondroprotection in the established rat meniscal tear model of OA [42,43]. In a recent Phase 2 trial, APPA was found to provide significant pain relief compared with placebo in a subset of patients with knee OA [44]. The combination of these compounds in APPA has potential to be an alternative to NSAIDs for managing the pain of OA and limiting disease progression.

Based on all these data and the potential of APPA to treat OA, the aim of this study was to analyze the capacity of APPA to modulate the inflammatory response induced by IL-1β in human articular chondrocytes and cartilage and explore the pathways affected.

## 2. Results

### 2.1. APPA Effect on Cell Proliferation

The analysis of cell proliferation in the presence of APPA showed that concentrations higher than 20 µg/mL decreased the cell viability less than 65% in 8 h (Figure 1A). The half maximal inhibitory concentration (IC_50_) showed at 24 h a value equal to 17.37 µg/mL (Figure 1B). The viability of chondrocytes in presence of 10 and 20 µg/mL APPA for 24 h reflected a viability higher than 85% (Figure 1C). 

### 2.2. APPA Effect over ROS Production in Human Chondrocytes

Total cytoplasmatic ROS and mitochondrial O_2_^−^ production were analyzed. LPS induced ROS production in chondrocytes (basal vs. LPS; 9.92 ± 0.70 vs. 22.81 ± 2.10, *p* = 0.007; and 5.59 ± 0.67 vs. 17.29 ± 0.78, *p* = 0.028). APPA decreased both types of ROS (cytoplasmatic and mitochondrial) induced by LPS (LPS vs. LPS + APPA; 22.81 ± 2.10 vs. 14.50 ± 2.52, *p* = 0.055; and 17.29 ± 0.78 vs. 9.28 ± 0. 1.72 = 0.028) (Figure 2A,B). 

The analysis of the detox system through the gene expression of nuclear factor erythroid 2 like 2 (*NFE2L2*), superoxide dismutase 2 (*SOD-2*) and inducible nitric oxide synthase (*iNOS*) was developed. Our results show that the gene levels of *NFE2L2*, *SOD-2* and *iNOS* increased in response to LPS or IL-1β. APPA caused a statistically significant reduction in *SOD-2* gene expression induced by IL-1β (IL-1β vs. IL-1β + APPA; 2.43 ± 0.86 vs. 1.11 ± 0.07, *p* = 0.007) (Figure 2C–E) (Table 1).

### 2.3. APPA Anti-Inflammatory Capacity

To evaluate the APPA anti-inflammatory capacity and its effect over the extra cellular matrix (ECM) degradation, the mRNA expression of Interleukin 6 (*IL-6*), Interleukin 8 (*IL-8*), Tumor Necrosis Factor alpha (*TNF-α*), Matrix Metallopeptidase 13 (*MMP-13*) and Matrix Metallopeptidase 3 (*MMP-3*) were analyzed. APPA 10 µg/mL ameliorated the gene expression of *IL-8* (0.423 ± 0.07, *p* = 0.02), *TNF-α* (0.454 ± 0.13, *p* = 0.02), *MMP-13* (0.513 ± 0.08 *p* = 0.02) and *MMP-3* (0.613 ± 0.09, *p* = 0.02), induced by IL-1β. The levels corresponding to *IL-6* did not show modulation in the presence of APPA (Figure 3).

### 2.4. APPA Effect in the Mitogensis

The effect of APPA over mitochondrial biogenesis and mitochondrial dynamic were evaluated through the mRNA expression of Peroxisome Proliferator-Activated Receptor Gamma Coactivator 1-Alpha (*PGC-1α*) and Mitofusin 2 (*Mfn2*). Chondrocytes incubated in the presence of APPA showed low levels of *PGC-1α* gene expression in comparison with IL-1β condition (0.94 ± 0.10 vs. 2.41 ± 0.79, *p* = 0.031) (Figure 4A). APPA did not modulate the gene expression of Mitofusin 2 (*Mfn2*) and Sirtuin 1 and 3 (*Sirt-1*, *Sirt-3*) (Figure 4B,C).

### 2.5. APPA Effect over the Human Articular Cartilage Degradation

To analyze the effects on proteoglycans (PG) loss, human cartilage explants were incubated with APPA for 24, 48, 72 and 96 h. No differences in Toluidine blue (TB) staining intensity were observed in the superficial layer and in the deep layer, However, significant effects were detected in the intermedial layer of the cartilage (Figure 5).

Glycosaminoglycan (GAGs) released into the culture medium was assessed in supernatants from 24 to 96 h (Figure 6A). The treatment of the cartilage explants with 100 µg/mL APPA reduced statistically significant the levels of s-GAG induced by IL-1β released into the supernatants at 24 h (60.28% ± 14.46, *p* = 0.028) and 48 h (80.31% ± 5.27, *p* = 0.028) (Figure 6B).

Taking into account the data described above, the incubation time increased at 16 days. Preliminary analysis reflected a tendency in the decrease in glycosaminoglycans released into the supernatant, and this reduction was maintained over time in comparison with the basal condition and with the IL-1β incubation (Figure 6C,D).

The gene expression of Matrix Metallopeptidase 3 (*MMP-3*) in human cartilage was evaluated, and values reflected that the incubation of cartilage in presence of APPA decreased *MMP-3* gene expression (at 24 h 60 µg/mL APPA 33.82% ± 7.77, *p* = 0.0004; 100 µg/mL APPA 26.45 ± 5.97, *p* = 0.0003) at 48 h 60 µg/mL APPA 30.24% ± 4.12, *p* = 0.0004; 100 µg/mL APPA 8.88 ± 2.94, *p* = 0.0002) (Figure 6E).

## 3. Discussion

The pathophysiology of OA is complex [45] with interplay between the joint tissues involving oxidative stress [46,47], chronic inflammation [48], cell senescence [49] and the production of tissue damaging enzymes, for example, metalloproteinases that digest cartilage [50]. In this work, we have shown that in human articular chondrocytes APPA decreased oxidative stress; reduced the expression of TNF-α and IL-8 (CXCL-8) but not IL-6; and reduced the expression of the collagen-degrading enzymes MMP-3 and MMP-13. The reduced expression of MMP-3 also occurred in cartilage explants. These effects translated into less proteoglycan loss in the intermedial layer of explants and lower levels of glycosaminoglycan release into the supernatant of the explants. Taken together these results indicate APPA has the potential to be an effective treatment of OA by mitigating several of the pathways leading to joint damage.

Oxidative stress has been reported as a key factor in the development of OA, resulting in increased inflammation and cartilage degradation as well as chondrocyte cell death [47,51,52]. In our study, APPA decreased both cytoplasmic and mitochondrial ROS production induced by LPS. APPA has been reported to reduce oxidative stress in experiments involving human neutrophils mainly through scavenging oxidative species rather than inhibiting their production. This study also reported that AP but not PA-reduced ROS [39].

We also evaluated the effects of APPA on the pathways activated in response to oxidative stress. When stimulated with LPS, human chondrocytes responded with a non-significant increase in the expression of the Nrf2 gene, *NFE2L2*, which became significant when the stimulus was instigated by IL-1β. APPA, however, had no effect in either scenario. In contrast to our experiments with human chondrocytes, it has been reported that APPA increased the gene expression of *Nrf2* in human neutrophils following stimulation with either TNFα or GM-CSF. However, in that study, higher concentrations of APPA were evaluated [39]. Similar results have also been reported for human skin fibroblasts stimulated with TGF-β1, where APPA upregulated both *Nrf2* and sirtuin 3 protein levels [53].

The levels of the superoxide dismutase gene, *SOD-2,* were increased by both LPS and IL-1β, and in both cases, that increase was inhibited by APPA. This counterintuitive finding could likely be best explained, in this experiment, by the early APPA inhibition of NADPH oxidase (NOX), which resulted in lower levels of ROS in the first instance. As a consequence, there was a reduced requirement for SOD enzyme release to prevent ROS-induced damage. We also found that there was a trend for APPA to reduce the IL-1β-stimulated increase in the expression of *iNOS*. The effects on the detox system reported here and by other researchers indicate that APPA protects against induced oxidative stress. AP is known to significantly block NADPH-oxidase activation, decreasing ROS production, and is also a strong ROS scavenger [30,33,34].

The beneficial effects provided by AP in animal models, involving oxidative and nitroxidative stress, include those for rheumatoid arthritis, diabetes mellitus type 2 and neurodegeneration and have been described in the literature [54,55,56]. Recently, PA was reported to inhibit mitochondrial-derived oxidative stress and the preservation of mitochondrial function in diabetic cardiomyopathy [57].

IL-1β, a key player in the pathogenesis of OA, induces the expression of various catabolic factors that contribute to cartilage degradation [58,59]. With this background and the connection between OA and low grade chronic inflammation through innate immunity pathways, it is interesting that APPA decreased the chondrocytes gene expression of the pro-inflammatory cytokine TNFα, the chemokine IL-8 (CXCL8) and the metalloproteinases MMP-3 and MMP-13 involved in extracellular matrix (ECM) degradation. Similar results with APPA and the expression of TNFα and IL-8 (CXCL8) were reported for activated human neutrophils stimulated with TNFα but unlike our experiments with chondrocytes, these authors also reported inhibition of IL-6 expression [39]. Our results with human chondrocytes were partially confirmed in cartilage tissue in that the reduced expression of MMP-3 and MMP-13 explained the effect of APPA on the levels of GAGs (glycose aminoglycans) released into the supernatant of the cartilage explants and the protection of proteoglycans in the intermediate layer. It has also been reported that APPA inhibits the enzymatic degradation of aggrecan in human cartilage explants stimulated with oncostatin M and TNF-α, indicating another pathway by which cartilage could be protected [60] Our data correspond with results described by other authors that AP inhibits the expression and the release of several inflammatory cytokines and matrix metalloproteinases [39], while PA downregulates activation, nuclear translocation and the DNA binding of NF-κB [61]. 

Mitochondrial homeostasis is preserved through the fine co-ordination between two opposing processes: the generation of new mitochondria, via mitochondrial biogenesis, and the removal of damaged mitochondria, via mitophagy [62]. PGC-1α, the main regulator of mitochondrial biogenesis, connects oxidative stress and mitochondrial metabolism with the inflammatory response and the metabolic syndrome [63,64]. It has been shown that increases in PGC-1α levels alter redox homeostasis in cells and exacerbate the inflammatory response, which is commonly accompanied by metabolic disturbances [63]. Our results show that APPA reduced PGC-1α induced by IL-1β. 

Our results presented here demonstrate that APPA impacts a number of the pathways implicated in the development and progression of OA (Figure 7), but a key question is whether these effects translate into clinical efficacy. Animal models have been widely used to assess the potential therapeutic efficacy of putative treatments for OA, which have included many induced and naturally occurring models in a range of species [65,66]. As described above, OA is a heterogenous disease, and likewise, the animal models vary and no single one can be considered to reflect the full spectrum of OA; there is no consensus on a model that reflects the range of phenotypes of human disease [67].

Evidence from two animal models of OA do, however, show that APPA protects against joint damage, namely in a study utilizing the widely used rat meniscal tear model [42] and a modified version of the OARSI scoring system [43] and also in a small study involving mongrel dogs, where joint damage was initiated by medial meniscal release and damage was evaluated using the canine OARSI score [68]. Importantly, APPA has also been shown to provide effective pain relief in naturally occurring canine OA [41,42]. In addition, in a Phase 2a double blind placebo controlled study in subjects with knee OA, APPA provided pain relief in a subset of patients who experienced nociplastic/neuropathic pain [44]. These positive results from animals and humans reflect the findings presented here, i.e., that APPA has been shown to impact pathways known to be involved in the pathogenesis of OA.

## 4. Materials and Methods

### 4.1. Drug Preparation

APPA was supplied by the AKL company in a prepared vial (2:7 ratio of AP:PA) and dissolved in Dimethyl Sulfoxide (DMSO) (Sigma-Aldrich, St. Louis, MO, USA) at a final concentration of 1 gr/mL, and serial dilutions were developed. 

### 4.2. Cartilage and Chondrocytes Isolation 

Written informed consent was obtained from all subjects involved in the study and approval was obtained from the local Ethics Committee of the Galician Health Administration (CEIC). All procedures were conducted according to the principles expressed in the Helsinki Declaration of 1975, as revised in 2000.

Tissue culture: Human OA articular cartilage was dissected from the femoral heads of five patients following total hip arthroplasty (female patients with a mean ± SD age of 72.2 ± 11.14 years) (Table 2). Cartilage samples were obtained using a 5 mm biopsy punch. Explants were equilibrated overnight in Dulbecco’s Modified Eagle’s Medium (DMEM) (Gibco, Grand Island, NY, USA) supplemented with 10% fetal bovine serum (FBS), penicillin (100 U/mL) and streptomycin (100 μg/mL) (Gibco) in a humidified 5% CO_2_ atmosphere at 37 °C. Each patient had at least four explants, and one per condition analyzed. One explant was used for the basal condition, while the other three had been stimulated on days 0 and 2 with IL-1β 10 ng/mL (Sigma-Aldrich) for 5 days (d). Then, the IL-1β was removed from the culture medium and one explant was stimulated with 60 µg/mL APPA and the other with 100 µg/mL APPA on days 0 and 2 for 4 d, and the time was increased in some experiments at 16 days. Each donor had: (1) basal condition, (2) 10 ng/mL IL-1β-5 d, (3) 60 µg/mL APPA and (4) 100 µg/mL APPA.

In all cases, the supernatant was collected and frozen at –80 °C, and the explants from each condition were removed and divided into two parts, one of which was fixed overnight in freshly prepared paraformaldehyde, dehydrated in an ethanol wash and incubated in Histo-Clear prior to embedding in paraffin wax for further histochemical analysis, and the other part was frozen in liquid nitrogen and kept at –80 °C for RNA extraction. 

Cell culture Chondrocytes. Chondrocytes were obtained from tissue donors and isolated from 29 patients following 2 total knee and 27 total hip arthroplasties (7 male and 22 female patients with a mean ± SD age of 75.89 ± 9.60 years) (Table 1) as previously described [69]. Chondrocytes were plated at a density of 5 × 10^4^ cells/well in Corning^®^ CellBIND^®^ multiwell 96 plates (MW 96) (Corning, NY, USA) or 1.8 × 10^5^ cells/well in MW 12 plates (Corning, NY, USA) for analysis. Cells were left to equilibrate overnight in DMEM at 5% of FBS in 5% CO_2_ at 37 °C. Cells were stimulated with 10 ng/mL IL-1β or 10 ng/mL LPS (Sigma-Aldrich).

### 4.3. Chondrocyte Viability (MTT) 

To test the APPA effect over chondrocyte viability, cells were grown under various experimental conditions. APPA was initially tested over a concentration range of 0 to 200 µg/mL (final concentration) for 8, 12 and 24 hours (h), and proliferation was determined using a CellTiter 96^®^ AQueous Non-Radioactive Cell Proliferation Assay (Promega, Madison, WI, USA), according to the manufacturer’s instructions. Absorbance was measured at 570 nm using a Tecan Infinite 200 microplate reader (Tecan, Männedorf, Switzerland).

### 4.4. Cytoplasmic Reactive Oxygen Species (ROS) and Mitochondrial Anion Superoxide (O_2_^−^) Production

ROS production was evaluated through flow cytometry (FACsCalibur, Becton Dickinson, Franklin Lakes, NJ, USA). Cells were cultured in MW 12 plates (Corning) (1.8 × 10^5^ cells/well). Cells were incubated in the presence of 10 ng/mL LPS for 2 h and then APPA (10 and 20 µg/mL) was added after another hour. To determine cytoplasmic ROS production, cells were incubated with 10 mM 2,7-dichlorodihydrofluorescein diacetate (DCFH-DA) (Sigma-Aldrich) for 30 min, and to measure mitochondrial anion superoxide (O_2_^−^) production, cells were treated with 5 mM MitoSOX^TM^ Red (Invitrogen, Carlsbad, CA, USA) for the same incubation time as DCFH-DA. 

Cells were harvested via trypsin release and resuspended in saline solution prior to analysis using flow cytometry. A total of 1 × 10^4^ cells per assay were measured. Data were analyzed using CellQuest Pro software (Becton Dickinson). Results were expressed as a median of fluorescence (AU).

### 4.5. Quantitative Real-Time PCR (qRT-PCR)

RNA extraction was achieved using TRIzol^®^ (Sigma-Aldrich), following the manufacturer’s protocol. A total of 0.5 mg of RNA was reverse transcribed into cDNA using Super Script VILO (Thermofisher Scientific, Waltham, MA, USA), following the manufacturer´s instructions. RT-PCR was developed in a LightCycler 480-II Instrument (Roche, Mannheim, Germany) using TaqMan Universal Master Mix (Roche). The analysis of the results was carried out using Qbase+ version 2.5 software (Biogazelle, Ghent, Belgium). Gene expression was calculated relative to the housekeeping gene *Glyceraldehyde-3-Phosphate Dehydrogenase* (GAPDH). Sequence primers (Table 3) and PCR conditions were described; a pre-incubation step at 95 °C for 5 min, followed by 40 cycles corresponding to amplification (denaturation at 95 °C for 10 seconds (s), annealing all the primers at 60 °C for 10 s with an extension at 72 °C for 1 s), and the last step was cooling at 4 °C for 5 min. Only a single acquisition mode was applied during the extension step in the amplification process.

### 4.6. Histological Analysis

Cartilage explants were sectioned at 4 μm thickness to show all layers of cartilage. Following slide de-waxing and rehydration, samples were stained with Toluidine blue (TB) (Merck, Darmstadt, Germany) dye. TB is a basic thiazine metachromatic dye with high affinity for acidic tissue components, and in cartilage, a purple color indicates a high amount of hyaluronic acid (HA), its most abundant metachromatic component.

All slides were then visualized using an Olympus Dx61 optical microscope (Olympus España S.A.U., Barcelona, Spain) and were subsequently imaged. The image quantification was developed using software ImageJ Fiji 1.54f [70].

### 4.7. Sulfated Glycosaminoglycan Assay (GAG)

The release of proteoglycan (PG), as well as sulfated glycosaminoglycan (s-GAG), was determined using a spectrophotometric assay, a commercial kit Blyscan-sulfated glycosaminoglycans assay, was used according to the manufacturer’s instructions (Biocolor, County Antrim, UK). Briefly, the reaction between s-GAG and dye produced a dye-GAG complex that precipitated out within 30 min. Then, an s-GAG-bound dye was recovered by adding dissociation reagent. Finally, the s-GAG content of the assayed samples was determined through the amount of dye recovered from the s-GAGs in the test sample by measuring absorbance at 656 nm, using a Tecan Infinite 200 microplate reader (Tecan), and was expressed as mg GAGs released into the supernatant. 

### 4.8. Statistical Analyses

Data are presented as the mean ± standard error of mean (SEM) from independent experiments with a minimum of four observations unless stated otherwise. Statistical analyses were performed using GraphPad Prism software version 8.0.0 (GraphPad, La Jolla, CA, USA). An unpaired Mann–Whitney test was used to evaluate differences between groups. Differences with *p* values of ≤0.05 were considered to be statistically significant.

### 4.9. Study Limitations

This work was developed using a limited number of human articular cartilage samples (in vitro model), and to validate these data, it could be interesting to increase the number of samples analyzed. All the samples used were obtained from hip joints, and in future studies, analyzing the APPA effect using samples from knee OA could be complementary. Most data described are related to gene expression; however, this technique has some limitations, and it could be interesting to develop future experiments focusing on proteomics analysis. The current work focused on the anti-inflammatory activity of APPA, and additional experiments are necessary to fully evaluate the possible role of APPA in other pathways related to OA pathogenesis.

## 5. Conclusions

In summary, our data show that in human chondrocytes and articular cartilage, APPA attenuated the expression of proinflammatory cytokines (IL-8 and TNF-α), catabolic genes (MMP-3 and MMP-13) and redox response genes (SOD-2 and iNOS), resulting in the protection of human articular cartilage. These findings overall indicate that APPA may be an effective treatment for OA by targeting several of the pathways involved in OA pathogenesis (Figure 7).

## Figures and Tables

**Figure 1 pharmaceuticals-17-00118-f001:**
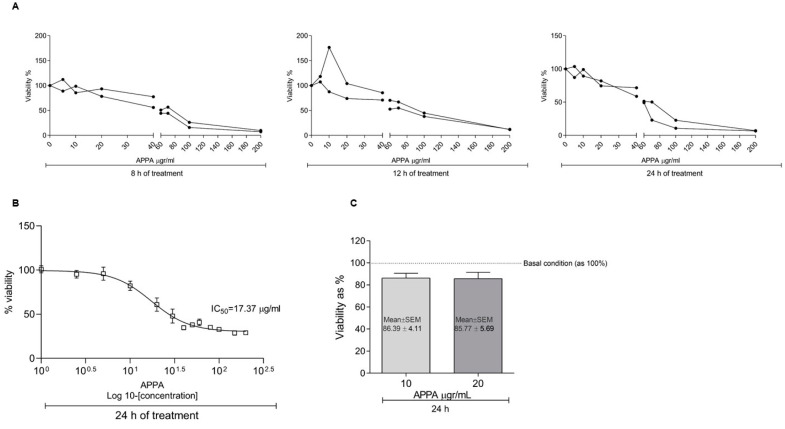
**Effect of APPA on cell viability.** (**A**) Cell viability (MTT assay) was measured after treatment with APPA at 0, 5, 10, 20, 40, 60, 70, 100 and 200 µg/mL concentrations for 8, 12 and 24 h. (**B**) Calculation of IC_50_ value; percentage of cell viability is provided on vertical axis and Log10^−^ (APPA concentration) on horizontal axis. (**C**) Chondrocytes viability after treatment with APPA at 10 and 20 µg/mL for 24 h. Results expressed as % of basal condition (100%). Data are represented as the mean ± standard error of mean (SEM); N = 5 performed in duplicate.

**Figure 2 pharmaceuticals-17-00118-f002:**
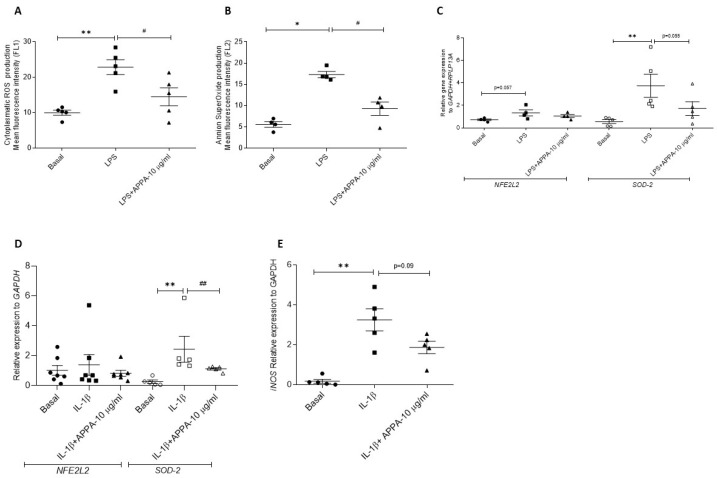
**APPA decreased reactive oxygen species (ROS) production induced in chondrocytes.** (**A**) Cytoplasmatic ROS production as measured through fluorescence intensity of 2,7-dichlorodihydrofluorescein diacetate (DCFH-DA). (**B**) Mitochondrial ROS production, as measured through fluorescence intensity of MitoSox^®^. (**C**) Relative gene expression *NEF2L2* (Nuclear Factor Erythroid 2-Related Factor 2) and *SOD-2* (Superoxide Dismutase 2). In this graph, the basal condition, 10 ng/mL LPS and LPS + 10 µg/mL APPA are represented. (**D**) Relative gene expression *NEF2L2* (Nuclear Factor Erythroid 2-Related Factor 2) and *SOD-2* (Superoxide Dismutase 2)—in this graph, the basal condition, 10 ng/mL IL-1β and IL-1β +10 µg/mL APPA are represented. (**E**) Relative gene expression *iNOS* (inducible Nitric Oxide Synthase)—in this graph, the basal condition, 10 ng/mL IL-1β and IL-1β + 10 µg/mL APPA are represented. Gene expressions were normalized to the housekeeping gene (*GAPDH*) in the y axis of graphs. Data were obtained at least from four independent donors and are presented as mean ± SEM. Analysis was carried out via the Mann–Whitney test * vs. basal condition and # vs. LPS or IL-1β condition (*, # *p* ≤ 0.05. **, ## *p* ≤ 0.01).

**Figure 3 pharmaceuticals-17-00118-f003:**
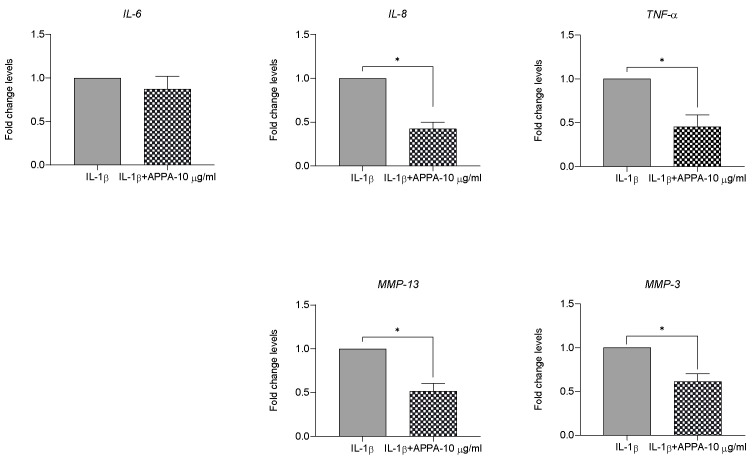
**APPA ameliorated the IL-1β-induced inflammatory response and ECM degradation in chondrocytes.** Effects of 10 µg/mL APPA on mRNA expression of Interleukin-6 (*IL-6*), *IL-8*, Tumor Necrosis Factor-α (*TNF-α*), Matrix Metallopeptidase-13 (*MMP-13*) and *MMP-3* are shown. Gene expressions are represented as fold change levels. All data were obtained from seven independent donors, each with two replicates. Values are presented as mean ± SEM and analyzed using a Mann–Whitney test (* *p* ≤ 0.05).

**Figure 4 pharmaceuticals-17-00118-f004:**
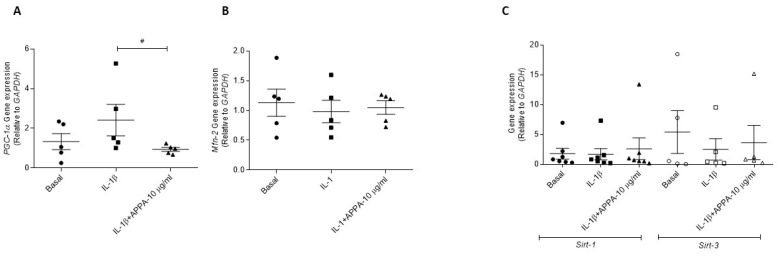
**APPA effect over mitogenesis**. Effect of 10 µg/mL APPA on (**A**) mRNA expression Peroxisome Proliferator-Activated Receptor Gamma Coactivator 1-Alpha, (*Pgc-1α*) (**B**) Mitofusin 2 (*Mfn2*) gene expression. (**C**) Gene expression of Sirtuin-1 (*Sirt-1*) and Sirtuin-3 (*Sirt-3*). Gene expressions were normalized to the housekeeping gene (*GAPDH*) in the y axis of graphs. All data were obtained from five independent donors, each with two replicates. Values are presented as mean ± SEM. Analyzed via Mann–Whitney test # vs. IL-1β condition (# *p* ≤ 0.05).

**Figure 5 pharmaceuticals-17-00118-f005:**
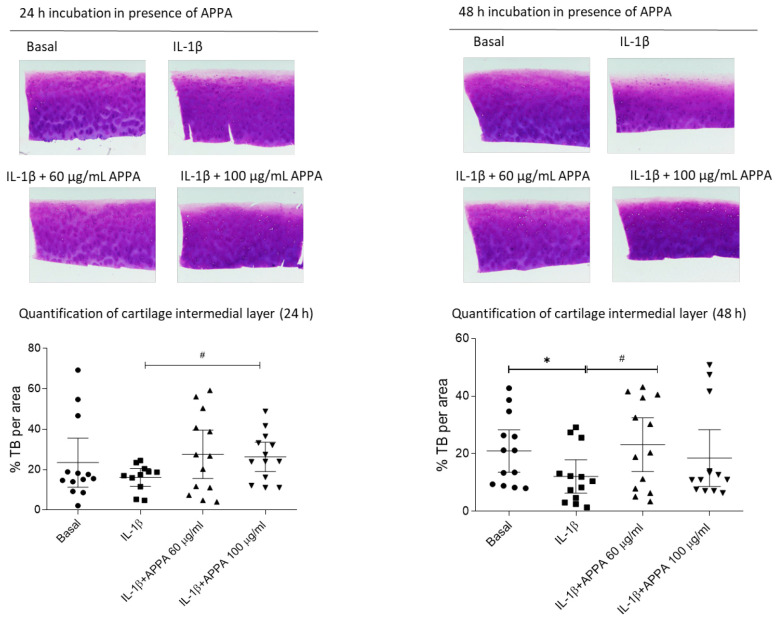
**Effect of APPA on the proteoglycans content**. (**Upper panels**) showed representative example of Toluidine blue (TB) staining. (**Lower panels**) showed the TB intermedial layer quantification of OA cartilage after 5 days of stimulation with IL-1β and then APPA was added for 24 h. and 48 h. All data were obtained from five independent donors. Values are presented as mean ± SEM and analyzed by Mann–Whitney test. * vs. basal condition and # vs. IL-1β condition (*, # *p* ≤ 0.05).

**Figure 6 pharmaceuticals-17-00118-f006:**
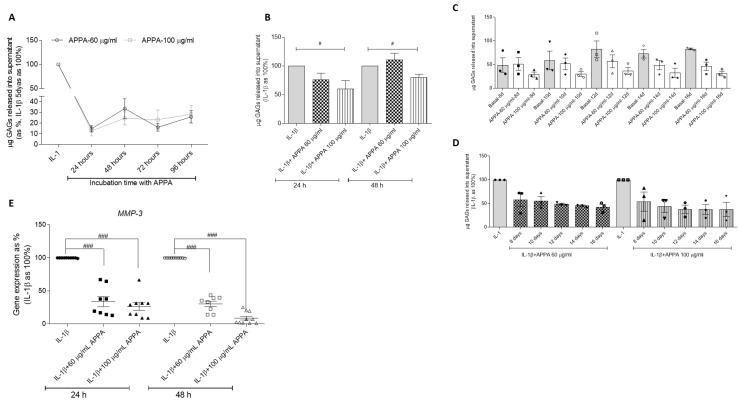
**APPA effect over Glycosaminoglycan (GAGs) release into the culture medium and *MMP-3* gene expression in human cartilage**. (**A**) µg GAGs released into the supernatant in OA cartilage for 24, 48, 72 and 96 h in presence of 60 (black line) and 100 (grey line) µg/mL APPA; IL-1β was considered to be 100%. (**B**) GAGs percentages released into the supernatant after 5 days of stimulation with IL-1β, and then APPA was added for 24 and 48 h. Data were obtained from five independent donors, each of them with two replicates. (**C**) µg of GAGs released into the supernatant in basal OA cartilage and after stimulation APPA at 60 and 100 µg/mL at 8, 10, 12, 14 and 16 days. (**D**) GAGs percentages released into the supernatant in OA cartilage after stimulation with IL-1β and then APPA at 60 and 100 µg/mL was added for 8, 10, 12, 14 and 16 days. These data (**C**,**D**) were obtained from three independent donors, each with two replicates. (**E**) Matrix Metallopeptidase 3 (*MMP-3*) gene expression was represented relative to the housekeeping gene *GAPDH.* Data were obtained from five independent donors, each with two replicates. Values are presented as mean ± SEM relative to IL-1β (as 100%) and analyzed via Mann–Whitney test and # vs. IL-1β condition (# *p* ≤ 0.05, ### *p* ≤ 0.0005).

**Figure 7 pharmaceuticals-17-00118-f007:**
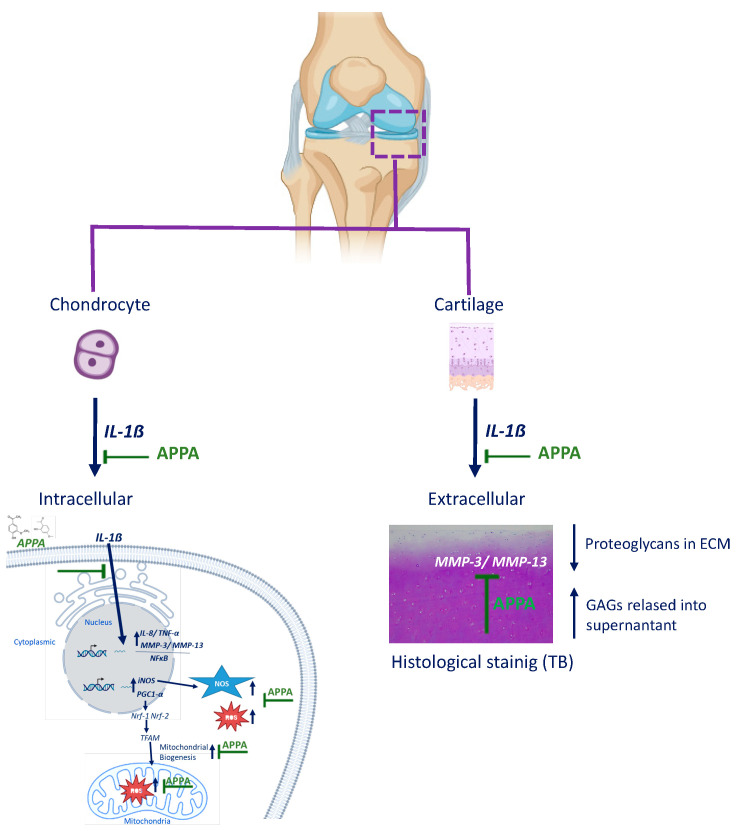
A schematic diagram summarizing the mechanism of APPA. ↑ increase ↓decrease.

**Table 1 pharmaceuticals-17-00118-t001:** **APPA effect on detox system of human articular chondrocytes**. APPA decreases gene expression levels induced by LPS or IL-1β. Data represented in the table corresponds to mean ± SEM; N = 4. * vs. Basal; # vs. positive stimuli (LPS or IL-1β) + APPA 10 µg/mL; ns, not significant.

	Gene Expression			Gene Expression	
	Basal	LPS	*p* Value	IL-1β+ APPA 10 µg/mL	*p* Value
** *NFE2L2* **	0.721 ± 0.07	1.33 ± 0.26	ns (*p* = 0.057)	1.05 ± 0.13	
** *SOD-2* **	0.566 ± 0.17	3.74 ± 1.03	** (0.007)	1.72 ± 0.60	ns (*p* = 0.055)
	**Gene Expression**			**Gene Expression**	
	**Basal**	**IL-1** **β**	***p* Value**	**IL-1** **β+ APPA 10 µg/mL**	***p* Value**
** *NFE2L2* **	1.01 ± 0.33	1.38 ± 0.69		0.82 ± 0.19	
** *SOD-2* **	0.255 ± 0.11	2.43 ± 0.86	** (0.007)	1.11 ± 0.07	## (0.007)
** *iNOS* **	0.16 ± 0.09	3.25 ± 0.55	** (0.007)	1.87 ± 0.31	ns (*p* = 0.092)
Mean ± SEM, *p* value (Mann Whitney)					
**, ## *p* ≤ 0.01					

**Table 2 pharmaceuticals-17-00118-t002:** Patient descriptions.

	Joint	Sex	Age	
			Mean	± SD
**Cartilage**	Hip	F (N = 5)	72.20	11.14
**Chondrocytes**	Knee	M (N = 2)	71.50	3.53
	Hip	M (N = 5)	76.33	9.81
		F (N = 22)	79.835	15.455

**Table 3 pharmaceuticals-17-00118-t003:** Sequence primers.

Gene Name	Symbol	Fw	Rv
*Nuclear Factor, Erythroid 2 Like 2*	*NFE2L2*	gcaacaggacattgagcaag	tggacttggaaccatggtagt
*Superoxide Dismutase 2*	*SOD-2*	ctggacaaacctcagcccta	tgatggcttccagcaactc
*Inducible Nitric Oxide Synthase*	*iNOS*	gctgccaagctgaaattga	gatagcgcttctggctcttg
*Glyceraldehyde-3-Phosphate Dehydrogenase*	*GAPDH*	gagtccactggcgtcttcac	gttcacacccatgacgaaca
*Interleukin 6*	*IL6*	gatgagtacaaaagtcctgatcca	ctgcagccactggttctgt
*Interleukin 8*	*IL8*	gagcactccataaggcacaaa	atggttccttccggtggt
*Tumor Necrosis Factor alpha*	*TNF-α*	gcaacaggacattgagcaag	tggacttggaaccatggtagt
*Matrix Metallopeptidase 13*	*MMP-13*	ccagtctccgaggagaaaca	aaaaacagctccgcatcaac
*Matrix Metallopeptidase 3*	*MMP-3*	caaaacatatttctttgtagaggacaa	ttcagctatttgcttgggaaa
*Peroxisome Proliferator-Activated Receptor Gamma Coactivator 1-Alpha*	*PGC1-α*	tgagagggccaagcaaag	ataaatcacacggcgctctt
*Mitofusin-2*	*Mfn2*	tcagctacactggctccaac	caaaggtcccagacagttcc
* Sirtuin-1*	*Sirt-1*	aaatgctggcctaatagagtgg	tggcaaaaacagatactgattacc
* Sirtuin-3*	*Sirt-3*	ccacctgctggattgtgac	ggagcctgtgcagaagtagc

## Data Availability

Data is contained within the article.

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
