# Peer review of "Anti-Inflammatory Activity of APPA (Apocynin and Paeonol) in Human Articular Chondrocytes"

_pharmaceuticals, 2024, doi:10.3390/ph17010118_

Round 1

Reviewer 1 Report

Comments and Suggestions for Authors

Review Comments

The title of the submitted manuscript to be reviewed: Anti-inflammatory activity of APPA (Apocynin and Paeonol) in human articular chondrocytes.

Dear author:

This manuscript presents an meaningful and well-conducted study concerning the anti-inflammatory activity of APPA, a combination of apocynin (AP) and paeonol (PA), in human articular chondrocytes. The paper addresses the urgent need for more effective treatments for Osteoarthritis (OA), a chronic joint disease characterized by cartilage loss, joint space reduction, and pain.

The authors explore the ability of APPA to modulate the inflammatory response in chondrocytes, focusing on the quantification of gene expression of proinflammatory cytokines, catabolic mediators, and redox response. The study also evaluates the effect of APPA on mitogenesis. Their results suggest that APPA can attenuate expression of IL-8, TNF-α, MMP-3, MMP-13, SOD-2, and iNOS, thereby protecting human articular cartilage, and also decrease PGC-1α gene expression induced by IL-1β.

While the research is solid and the content substantial, there are two suggestions for improvement:

1. A schematic diagram summarizing the mechanism of the “Anti-inflammatory activity of APPA (Apocynin and Paeonol) in human articular chondrocytes” would be beneficial. This would provide readers with a more intuitive understanding of the content of the article.

2. In Figure 5, it would be helpful to include images of Toluidine Blue (TB) staining in the superficial, middle, and deep layers of the cartilage. Displaying the intensity characteristics of Toluidine Blue (TB) staining at different cartilage depths would give readers a more intuitive understanding of the staining’s properties.

In conclusion, this paper makes a valuable contribution to the understanding of APPA’s potential as an OA treatment. The suggestions above aim to enhance the clarity and overall presentation of the already commendable work. Looking forward to the revised manuscript.

In recent years, the research and application of small molecule drugs to treat osteoarthritis by regulating related pathways have also been reported in the literature, especially the research progress of related natural drugs and derivatives, which have the potential to effectively inhibit symptoms and regulate homeostasis. It is recommended to quote in the appropriate place, such as:

1. Fang C, Guo JW, Wang YJ, Li XQ, Zhang H, Cui J, Hu Y, Jing YY, Chen X, Su JC. Diterbutyl phthalate attenuates osteoarthritis in ACLT mice via suppressing ERK/c-fos/NFATc1 pathway, and subsequently inhibiting subchondral osteoclast fusion. Acta Pharmacol Sin. 2022 May;43(5):1299-1310. doi: 10.1038/s41401-021-00747-9. Epub 2021 Aug 11

2. Zhou D, Zhang H, Xue X, Tao Y, Wang S, Ren X, Su J. Safety Evaluation of Natural Drugs in Chronic Skeletal Disorders: A Literature Review of Clinical Trials in the Past 20 years. Front Pharmacol. 2022 Jan 13;12:801287. doi: 10.3389/fphar.2021.801287.

3. Wang, W., Mai, H., Xu, H. et al. 4,8-Dicarboxyl-8,9-iridoid-1-glycoside inhibits apoptosis in human osteoarthritis chondrocytes via enhanced c-MYC-mediated cholesterol metabolism in vitro. Arthritis Res Ther 25, 240 (2023).

4. Wang, L., Xu, H., Li, X. et al. Cucurbitacin E reduces IL-1β-induced inflammation and cartilage degeneration by inhibiting the PI3K/Akt pathway in osteoarthritic chondrocytes. J Transl Med 21, 880 (2023). 

5. Wang, Z., Wang, X., Liu, L. et al. Fructose-bisphosphatase1 (FBP1) alleviates experimental osteoarthritis by regulating Protein crumbs homolog 3 (CRB3). Arthritis Res Ther 25, 235 (2023). 

6. Liang, M., et al., Replenishing decoy extracellular vesicles inhibits phenotype remodeling of  tissue-resident cells in inflammation-driven arthritis. Cell Rep Med, 2023. 4(10): p. 101228.

Author Response

Dear reviewer-1

Thank you for your comments. We include a schematic diagram to summarize our data at the end of the manuscript following your recommendation under the name of figure 7. In this figure we want to provide a more intuitive understanding of the contents of all data described in our article.

Figure 5 was modified including upper panels with representative images of TB staining and the quantification of intermedia layer of the cartilage showed in the lower panels.

Following your last recommendation, we included this paragraph in the introduction of the article between line 64 to 66 “In recent years, the application of small molecule drugs to treat osteoarthritis, especially the research progress of related natural drugs or derivatives, which have the potential to inhibit symptoms and/or regulate homeostasis of cartilage, have also been reported in the literature (we included the reference that you suggested to us)”

Reviewer 2 Report

Comments and Suggestions for Authors

APPA has been previously identified to treat OA. In the current study Mercedes et. al tried to evaluate the anti-inflammatory potential of APPA in the human articular chondrocytes. Few aspects of this study have already been established, however, this study specifically investigated the cytokine responses in terms of its anti-inflammatory potential to target human articular chondrocytes. The study has merits, however, needs to be more informative.

Line 21: Capacity may be replaced to efficacy

Line 22-24: Concise by joining the sentences

Line 64-68: Previous studies explaining the anti-inflammatory potential of APPA may be included in the manuscript (PMID: 32383062)

Line 64-76: Established animal models of OA needs to be included in the beginning of the paragraph (PMID: 32062692 (latest review), 30600472 (articular damage model) and 36829562 (OA-induction models). Line 64-76 may be made into a single paragraph.

Line 52-63 may also be made into a single paragraph by paraphrasing.

Line 79: The sentence needs to be corrected. Abbreviate h in the first instance.

Line 80: the sentence may be corrected, IC50 is repeated.

Line 90: The sentence may be included in the discussion section.

Line 91: Positive stimuli??

Line 95, 96: Genetic expression?? May be corrected everywhere

Line 96: Data may be replaced to Our results shows

Figures: The gene expressions must be represented as fold change (fold increase or increase), not as percentage. Normalize the expression with the housekeeping gene used. Also mention ‘normalized to (name of house keeping gene)’ in the y axis of graphs.

Materials and Method: The primer design and q-RT-PCR condition must be provided.

Line 264: correct as ‘reverse transcribed’

Line 270: Histological analysis method was explained. Include the results in the manuscript.

Line 289: The study limitations may be included.

Author Response

Dear reviewer-2

Thank you for your comments and suggestions

We included all your suggestion in the introduction hoping that this part of the article could increase its informative level. Here we described only one of your suggestions that needs some comments:

Line 64-68: Previous studies explaining the anti-inflammatory potential of APPA may be included in the manuscript (PMID: 32383062).

Response: To clarify this point, we include in line 72 the next sentence “Anti-inflammatory potential of APPA has been described using neutrophils” with the reference at the end.

Line 64-76: Established animal models of OA needs to be included in the beginning of the paragraph (PMID:3 2062692 (latest review), 30600472 (articular damage model) and 36829562 (OA-induction models). Line 64-76 may be made into a single paragraph.

Response: Data described in the article were obtained from in vitro model the paragraph where the effect of APPA using OA models was included only in the introduction. The OA model described in this section was only spontaneous OA in dog, the articles cited in this part did not use models with induced OA. For this reason, we not include in the introduction a paragraph described the OA models, because the articles described here no include OA model, the authors described the APPA affect in animals with spontaneous OA no with induced OA. However, following your instructions we put line 64 to line 73 in a single paragraph and the last part (line 74 to 76) in an independent paragraph because is a small conclusion of data described before.

Line 91: Positive stimuli??

Response: We change this expression by LPS

Figures: The gene expressions must be represented as fold change (fold increase or increase), not as percentage. Normalize the expression with the housekeeping gene used. Also mention ‘normalized to (name of house keeping gene)’ in the y axis of graphs.

Response: The gene expression was represented in different ways in function of the conditions evaluated in each experiment. In figures 2 and 4 we described the gene expression relative to housekeeping (GAPDH, we include the gene name in the y axis). We did not represent the gene expression as fold change because we want showed the row data to do the comparison between the three conditions and no relative to one of them. In the case of figure 3 we showed the data as percentage because we want show the capacity to APPA in decrease the gene expression in relation to the IL-1ẞ condition, give to this condition the maximum modulation of gene expression. In figure 2 and 4 we include this sentence to clarify the interpretation of data “Gene expressions were normalized to the housekeeping gene (GAPDH) in the y axis of graphs

Materials and Method: The primer design and q-RT-PCR condition must be provided.

Response: Following your suggestion, we include a table with primer sequence (Table 3). The q-RT-PCR conditions used were the normal conditions described by Roche, this is the reason to not include in the manuscript, but we described for you here “The reaction mixture included: 5 µL of TaqMan Universal Master Mix (Roche), 0,7 µM of each primer, 0,1 µL of each probe and water to obtain a final volume of 10 µL. In each tube we used 2,5 µL of cDNA. The amplification process using the LightCycler® 480 II instrument (Roche) followed the methodolgy for TaqMan probe described by Roche

Line 270: Histological analysis method was explained. Include the results in the manuscript.

Response: Following your suggestion, we include some histological images relative to TB staining. Figure 5 was modified including upper panels with representative images of TB staining and the quantification of intermedia layer of the cartilage showed in the lower panels.

Line 289: The study limitations may be included.

Response: Following your recommendation, we included a small paragraph describing the limitations of the study before conclusions point “This work was developed using human articular samples (in vitro model) and to validate this data could be interested increase the number of sample analyzed. All the sample used were obtained from hip and study the APPA effect using sample from knee OA could be complementary. Most of data described are related with gene expression, with the own limitations that have the used of this technique for this could be interested develop more experiments focusing proteomics analysis. This work is focused in the anti-inflammatory activity of APPA, next experiments could necessary to evaluate the APPA capacity over other pathways related with the OA process

Round 2

Reviewer 2 Report

Comments and Suggestions for Authors

Thank you for the responses from the authors.

Most of the comments are ok; however, I still need clarification on few important points.

Discussing the experimental OA models including the OA induction model is required in the manuscript. I suggest these models would be the easy fit to explore the future possibility on the therapeutic aspects of APPA in OA progression.

Regarding the gene expressions explained as percentage, for example Fig. 3, the authors compared the different groups. This data can be best explained in fold times. If in percentage, what makes it different from others? In case of representation in percentage, how could the authors make sure that equal quantity of RNA has been used? To make sure of equal quantity of RNA used, normalization with a housekeeping gene is necessary.

Refer the response from authors, the authors used 2.5μL or μg of total cDNA? The PCR condition needs to be included in the manuscript for other researchers to follow. Otherwise, cite if the methodology is adopted from somewhere.

Author Response

Dear Reviewer 2, Thank you for your new comments

  • Discussing the experimental OA models including the OA induction model is required in the manuscript. I suggest these models would be the easy fit to explore the future possibility on the therapeutic aspects of APPA in OA progression.

Response: Following your suggestion we introduced at the end of the discussion (line 230 to 243) this paragraph; where we described the experimental OA models related to APPA. “Our results presented here demonstrate that APPA impacts on a number of the pathways implicated in the development and progression of OA, but a key question is whether these effects translate into clinical efficacy. Animal models have been widely used to assess the potential therapeutic efficacy of putative treatments for OA, which have included many induced and naturally occurring models in a range of species (65, 66). As described above, OA is a heterogenous disease and likewise the animal models vary and no single one can be considered to reflect the full spectrum of OA and there is no consensus on a model that reflects the range of phenotypes of human disease (67).

Evidence from 2 animal models of OA do however show that APPA protects against joint damage in the widely used rat meniscal tear model (42) using a modified version of the OARSI scoring system (43), and also in a small study involving mongrel dogs where joint damage was initiated by medial meniscal release; damage was evaluated using the canine OARSI score (68). Importantly APPA has also been shown to provide effective pain relief in naturally occurring canine OA (41, 42). In addition, in a Phase 2a double blind placebo controlled study in subjects with knee OA, APPA provide pain relief in a subset of patients who experienced nociplastic/neuropathic pain (44). These positive results from animals and humans reflect the findings presented here where APPA has been shown to impact on pathways known to be involved in the pathogenesis of OA.”

  • Regarding the gene expressions explained as percentage, for example Fig. 3, the authors compared the different groups. This data can be best explained in fold times. If in percentage, what makes it different from others? In case of representation in percentage, how could the authors make sure that equal quantity of RNA has been used? To make sure of equal quantity of RNA used, normalization with a housekeeping gene is necessary.

Response: We always use the same quantity of RNA in all the assays. We always used a housekeeping and all genes expressions were normalized to the housekeeping gene (GAPDH). Following your instructions, we change the figure 3 represented the data corresponding to gene expression as fold change levels.

  • Refer the response from authors, the authors used 2.5μL or μg of total cDNA? The PCR condition needs to be included in the manuscript for other researchers to follow. Otherwise, cite if the methodology is adopted from somewhere.

Response: We used 2.5 µl of cDNA (obtained from the 500 ng of RNA retro-transcription) in a total volume of 20 µl for each reaction (at the we used 62.5 ng of cDNA per sample). Followed your suggestion, between line 295 to 298, we included a paragraph describing the PCR condition “PCR conditions were pre-incubation step at 95ºC for 5 min following by 40 cycles corresponding to amplification (denaturation at 95ºC for 10 seconds (sec), annealing for all the primers at 60ºC for 10 sec and an extension at 72ºC for 1 sec) and the last step was a cooling at 4ºC for 5min. A single acquisition mode only was applied during the extension step in the amplification process”.

Round 3

Reviewer 2 Report

Comments and Suggestions for Authors

I agree with the responses and revised manuscript. Thank you.